
# New Study on the 1941 Gloria Fault Earthquake and Tsunami

Maria Ana Baptista[1,2], Jorge Miguel Miranda[2,3], Josep Batlló[4], Filipe Lisboa[3], Joaquim Luis[2,5], Ramon Maciá[6]

5   1 Instituto Superior de Engenharia de Lisboa, Instituto Politécnico de Lisboa, Lisboa, Portugal

2 Instituto Dom Luiz, Universidade de Lisboa, Lisboa, Portugal

3 Instituto Português do Mar e da Atmosfera, Lisboa, Portugal

Institut Cartográfic i Geológic de Catalunya, Barcelona, Spain

Universidade do Algarve, Faro, Portugal

6 Dept. de Matemàtiques, Universitat Politècnica de Catalunya, Barcelona, Spain

*Correspondence to: Maria Ana Baptista (mavbaptista@gmail.com)*

**Abstract.** The M~8.3-8.4 25th November 1941 was one of the largest submarine strike-slip earthquakes
ever recorded in the North East (NE) Atlantic basin. This event occurred along the Eurasia-Nubia plate
boundary between the Azores and the Strait of Gibraltar. After the earthquake, the tide stations in the NE
Atlantic recorded a small tsunami with maximum amplitudes of 40 cm peak to throw in Azores and Madeira
islands. In this study, we present a re-evaluation of the earthquake epicentre location using seismological
data not included in previous studies. We invert the tsunami travel times to obtain a preliminary tsunami
source location using a backward ray tracing (BRT) technique. We invert the tsunami waveforms to infer
the initial sea surface displacement using Empirical Green Functions without prior assumptions on the
geometry of the source. The results of the BRT simulation locate the tsunami source quite close to the new
epicentre. This fact suggests that the co-seismic deformation of the earthquake induced the tsunami. The
waveform inversion of tsunami data favours the conclusion that the earthquake ruptured approximately 160
km segment of the plate boundary, in the eastern section of the Gloria Fault between $-20.249^0$E and $-18.630^0$E. The results presented here contribute to the evaluation of tsunami hazard in the North East
Atlantic basin

## 1. Introduction

The western segment of the Eurasian-Nubian plate boundary extends from Mid-Atlantic Ridge in the
Azores towards the Strait of Gibraltar. Between -24ºE and -20ºE, the plate boundary is supposed to follow
a prominent morphological feature, the Gloria Fault firstly mapped by Laughton et al. (1972) (see figure 1
for location). The development of Gloria Fault as a segment of the Eurasia-Nubia plate boundary occurred
~27Myr ago (Luis and Miranda, 2008). The relative interplate motion is slow ~ 5 mm/year (Fernandes et
al., 2003) with unusual seismic activity. In spite of this, a few large earthquakes have been located close to
this plate boundary. The predominant focal mechanism is strike–slip (Buforn et al., 1998, 2004; Argus et
al., 1989). Among these events, are the 25th November 1941 (Udias et al., 1976, Lynnes and Ruff, 1985),





the 9th June 1969 (Argus et al., 1989), and the 26th May 1975 (Buforn et al., 1988; Argus et al., 1989; Kaabouben et al., 2008) and the 17[th] October 1983 (Argus et al., 1989). Some of these earthquakes
generated tsunamis, described in historical documents and recorded by the tide stations, namely 31[st] March 1761, the 25[th] November 1941 and the 26[th] May 1975 (Baptista et al., 2006; Kaabouben et al., 2008; Baptista and Miranda, 2009).

In this study, we present a re-analysis of the seismic data of the 25 November 1941 event and the first comprehensive analysis of the associated tsunami. We use a new set of old seismograms to re-evaluate the
position of the epicenter and the computation of the seismic moment. We digitized, de-tided and filtered all tide records available in the North East Atlantic basin. The use of seismic and tsunami data provides a better understanding of this earthquake and tsunami.

**2. The 25th November 1941 Earthquake**

The 25th November 1941 earthquake occurred at 18:03:54 (TUC). Madeira, Azores and western Portugal recorded the strongest shaking – VI (MSK). The earthquake impacted the neighboring countries: Spain and Morocco. The studies by Debrach (1946), Di Filippo (1949) and Moreira (1968) present macroseimic data analyses.

Antunes (1944) presented the first epicenter location, using only data from the Portuguese seismic network,
at -18.9ºE, 38.7ºN. Gutenberg and Richter (1949) relocated the epicenter at -18.5ºE, $37.50^{0}$N and computed the earthquake magnitude as Ms=8.2. Later, Lynnes and Ruff (1985) using a master-event technique relative to a better characterized event (26th May 1975) relocated the epicenter at $-19.1^{0}$E, $37.6^{0}$N. Udias et al. (1976) computed the earthquake magnitude as 8.3.

In this study, we present a relocation of the earthquake source using the phases published by the ISC
bulletin complemented with readings from nearby stations AVE in Morocco ($-7.413^{30}$E, $33.2961^{0}$N), FBR in Spain ($2.1239^{0}$E, 41.41840N) and MAL in Spain ($-4.4292^{0}$E, $36.762^{0}$N). To compute the hypocenter, we used "Hypocenter" code running under SEISAN environment (Ottemoler et al., 2011).

Aditionally, we tested two velocity models to compute the epicenter. The IASPEI91 velocity model (Kennett et al., 1991) locates the epicenter at $-18.965^{0}$E, $37.344^{0}$N. The regional velocity model by Carrilho
et al. (2004) for the Portuguese area locates the epicenter at $-19.038^{0}$E, $37.405^{0}$N, which is our preferred solution. The focal depths given by these models are 12km (rms=2.5s) and 5.3km (rms=2.2s) respectively.

We used the (Dineva et al., 2002) approach to compute the seismic moment from body wave spectra, using twenty six seismograms from fourteen seismic stations. Using P waves from thirteen stations we obtain a seismic moment of 3.96 1021 Nm, while using S waves from seven stations the seismic moment gives 2.96
$x10^{21}$ Nm. These seismic moment values correspond to a moment magnitude of 8.3±0.2 and 8.2±0.3 respectively . To re-evaluate the focal mechanism, we used the polarities from seventy two seismograms. The parameters of the focal mechanism are strike $76^{0}$, dip $88^{0}$ and rake angle $-161^{0}$. The focal mechanism is depicted in figure 1

**3. Tsunami data analysis**





After the earthquake, the tide stations in Portugal mainland, Morocco, Madeira and Azores Islands (see figure 1 for locations) recorded a small tsunami. In UK, Newlyn (-5.5$^c$E, 50.1$^0$N) tide station (not shown in figure 1) recorded the tsunami. Ponta Delgada tide station, in the Azores, recorded the maximum amplitude of 0.2 m. The Portuguese press reported tsunami observations close to Lisbon, and the overtopping of

shallow areas close to Oporto (Leixões) (see Baptista and Miranda, 2009 for details). Haslett and Bryant (2008) describe tsunami observations along the English Channel at Newlyn (Cornwall, United Kingdom) and Le Havre (France). Debrach (1946) and Rothé (1951) report damage to the submarine cables Brest–Casablanca and Brest–Dakar. In Spain, the Bulletin of the Seismic Observatory in Almeria reports tsunami observations wave in the Canary Islands but the tide record of Tenerife is unreadable.

In this study, we selected the records of seven tide stations presented in Table I. We digitized the original paper records except those of Casablanca and Essaouira (Morocco). Debrach (1946) presents drawings of these tide-records: Casablanca-Petite Darse, Casablanca-Jetee-Transversale and Essaouira. We could not find the original records of these stations. Instead, we had to rely on the drawings reproduced in Debrach (1946). We discarded the station Casablanca-Petite-Darse because the position of the tide gauge is

uncertain. In this study, we use Casablanca-Jetee-Transversale, here named Casablanca. We did not use the record of Newlyn because of the quality of the record.

All digitized records were linearly interpolated into a one-minute time step. For de-tiding, we used a least squares algorithm to fit the interpolated data into polynomials whose degrees guaranteed an adjusted R-square parameter above 0.995. To further ensure the goodness of our de-tiding we applied a band-pass

filter. The parameters of the filter are: end of the first stop band at 91 min, beginning of first pass-band at 89 min, end of the pass-band at 5 min and beginning of the second stop band at 2 min. The filter's magnitude response is of 30 dB attenuation in the first stop-band, a pass-band ripple of 1dB and a second stop-band attenuation of 40 dB. We used Morlet's continuous wavelet analysis. Wavelet analysis has the advantage of providing information when a particular wave component is present, even if it appears only once in the

time series. Hence, it can be better suited to the analysis of transient signals like tsunami waves. In this study, we present the results of wavelet analysis (equation 1), using the Morlet mother function (equation 2):

$$W_{\Psi|f}(a,b) = \int_{-\infty}^{+\infty} \frac{1}{\sqrt{a}} \Psi^* \left(\frac{t-b}{a}\right) f(t) \partial t \quad a, b \in \mathcal{R}, a > 0 \tag{1}$$

The Morlet mother function is given by a plane wave modulated by a Gaussian:

$$\Psi_0(\eta) = \pi^{-1/4} e^{i\omega_0\eta} e^{-\eta^2/2} \tag{2}$$

Where $\omega_0$ is the non-dimensional frequency, here taken to be 6 to satisfy the admissibility condition.

Figure 2. depicts the tsunami signals and the results of wavelet analysis. Madeira station presents the smallest dominant period close to 6 minutes. Ponta Delgada, Cascais and Lagos present a dominant period close to 12 minutes. Leixões presents a dominant period close to 20 minutes but this period is present in

the spectrum well before the arrival of the tsunami - in this station the signal to noise ratio is low, when compared with the other records. Casablanca presents a 22.5 minute dominant period. In the case of Essaouira, the dominant period is 45 minutes.





### 4. Preliminary location of the tsunami source

We used the travel time of the first wave arrival at each station to compute a preliminary location of the tsunami source. To do this, we used the Backward Ray Tracing (BRT) technique (Gjevik et al., 1997). We defined a set of points of interest (POIs) one per tide station. The position of each POI is the grid node closest to the actual location of the tide station, at a depth not less than 10 meters to avoid strong non-linear effects. Table I shows the location and depth of POIs used in this study. We back propagated the wave

fronts from each POI using the algorithm implemented in the Mirone suite (Luis, 2007). The location of the tsunami source is the minimum of the averaged travel time square errors (in the least squares sense). The spatial distribution of the misfit is given by:

$$\varepsilon^2(x, y) = \frac{\sum_{i=1}^{n}(t_i^o - t_i^p)}{n} \tag{3}$$

where $t_i^o$ and $t_i^p$ are the observed and predicted travel time for each POI and n is the total number of POIs.

To compute the BRT solution, we used the tsunami travel times shown in Table 1 and a bathymetric grid with a horizontal resolution of 0.1°. In figure 3, we present the minimum of the averaged travel time error, $\varepsilon(x, y) = 7.2$ minutes at $-19.0^0$E, $37.35^0$N. This location is 10 km to the SE of the seismic epicenter - 19.038ºE, 37.405 ºN, presented in section 2 (see figure 3).

        The BRT method has limitations inherent to the linear shallow water approximation implying an

overestimation of the speed in shallow waters. Nevertheless, in the case of the 1941 tsunami, the good azimuthal coverage ensures that these limitations are partially averaged out.

### 4. Tsunami waveform inversion

### 4.2 Inversion methodology

To estimate the initial sea surface displacement (tsunami source) we need to invert the tsunami waveforms. Moreover, for tsunamis generated by earthquakes, we assume that the initial sea surface displacement mimics the elastic deformation of the seafloor thus providing information on the earthquake focal mechanism.

        The problem of tsunami waveform inversion without assuming a fault model was firstly addressed by Aida

(1972, in Satake, 1987). Since then, many studies focused on the use of waveform inversion methods to estimate the tsunami source. These studies can be broadly divided in two categories: with a priori assumptions on the fault model (e.g. Satake, 1987 and 1993; Hirata et al. 2003; Titov et al. 2005), and without a priori assumptions on the the source namely Baba et al. (2005), Satake et al. (2005), Tsushima et al. (2009), Wu and Ho (2011), and Yasuda and Mase (2013).

Here, we use the method proposed by Tsushima et al. (2009), Koike et al. (2011) and Miranda et al. (2014) based on the use of Empirical Green Functions to efficiently perform the linear shallow water forward problem, where the synthesis of the tsunami waveform $\eta_m(t_t)$ at the m point of interest along the coast for the time span $t_t$ is given by a superposition principle:

$$\eta_m(t_t) = \sum_{i=1}^{nl} \sum_{j=1}^{nc} G_{ij}^m(t_t) h_{ij} \tag{4}$$





where (nl x nc) are the set of unit sources and $h_{ij}$ is the amplitude of the initial water displacement attributed

to the ij unit source. $G_{ij}^m(t_t)$ is the response to the unitary source of the ith element of the sea surface at the

jth location and m is the total number of unitary sources. The waveform inversion is simply obtained by

solving:

$$\begin{bmatrix} G_{11}^1(t_1) \ G_{12}^1(t_1) \ G_{13}^1(t_1) \ ... \ G_{nl\,nc}^1(t_1) \\ ... \\ G_{11}^1(t_t) \ G_{12}^1(t_t) \ G_{13}^1(t_t) \ ... \ G_{nl\,nc}^1(t_t) \\ ... \\ G_{nl\,nc}^m(t_1) G_{12}^m(t_1) G_{13}^m(t_1) \ ... \ G_{nl\,nc}^m(t_1) \\ ... \\ G_{nl\,nc}^m(t_t) \ G_{12}^m(t_t) \ G_{13}^m(t_t) \ ... \ G_{nl\,nc}^m(t_t) \end{bmatrix} \begin{bmatrix} h_{11} \\ h_{12} \\ h_{13} \\ ... \\ h_{nl\,nc} \end{bmatrix} = \begin{bmatrix} \eta_1(t_1) \\ ... \\ \eta_1(t_t) \\ ... \\ \eta_m(t_1) \\ ... \\ \eta_m(t_t) \end{bmatrix}$$    (5)

We incorporate the suggestion by Tsushima et al. (2009) defining an influence area around the minimum

of the BRT. The initial water displacement is forced to become null as the distance to the minimum

increases, therefore allowing the computation of a spatially compact solution. To do this, we added to (5)

additional equations of the form:

$$[0 \ 0 \ 0 \ 0 \ ... \ q \ ... \ 0] \begin{bmatrix} h_{11} \\ h_{12} \\ h_{13} \\ ... \\ h_{nl\,nc} \end{bmatrix} = [0]$$    (6)

where q is a positive number given by:

$$q(r) = 10 * (\frac{r}{r_{max}} - 1)$$    (7)

and r>rmax is the distance between each parameter (unit source) and the minimum of BRT. rmax is a cut-

off distance after which the initial water displacement converges to zero. Equations (5) and (6) form a

mixed-determined linear inversion problem (Menke, 2012) which can be solved in the sense of the least

squares weighted minimum norm, by;

$$h^{est} = [G^T W_e G + p^2 D^T D]^{-1} G^T W_e \eta$$    (8)

In equation (8), D is the first order derivative operator, We is a diagonal matrix that weights the likelihood

that we attribute to the different observations. Each observation corresponds to a single measurement made

by a tide gauge. Different values of p correspond to various levels of smoothness of the final solution.


### 4.2 Application to the 25 November 1941 event

In this work, the study area is shown in figures 1 and 2, which limits are -26.28$^0$E to -5.020E and 31.08$^0$N

to 42.65$^0$N. The Green Functions Database covers the area defined by -22.5$^0$E to -17.4$^0$E and 35.5$^0$N to

38.6$^0$N. This area has a total of 51 x 31 unit prisms. The dimensions of the unitary prisms are approximately

0.12º x 0.12º. The time span for the calculation was fixed as 180 min, and the time step as 1 min.

Experience shows that giving the same weight to all data points is misleading as later arriving waves carry

information resulting from reflections and interference, which are very hard to simulate with the inversion

procedure. Here, we used a simple strategy of attributing a larger weight to the first incoming wave. We

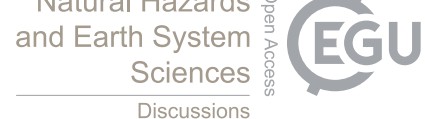



used only two different weights (eq. 8) for the data set. Data between the origin time of the earthquake and the first tsunami wave weighted 100 times more than the remaining data values, except for the Essaouira data.

Using the results of the BRT, we selected the area within the 10-minute contour (corresponding to an average travel time error of less than 10 minutes) as the influence area (see figure 3). We performed several tests to assess the best choice of the parameters (p, rmax) that describe the relative quality of the data, the

smoothness in the final solution and the size of the influence area, respectively. Parameter p is non-dimensional while r is measured in grid units, and corrected for the latitude.

In figure 4, we show a "chessboard" synthetic source, with alternating +1m/-1m displacements, to study the resolving power of the inversion procedure. The test reproduces the geometry of the inversion procedure (same location of the POIs, and identical values of p, and rmax). The results presented in figure 4 show that

the inversion algorithm recovers most of the "chessboard" information inside the influence area. Different values of rmax do not change qualitatively the results for the inner area. Nevertheless, the value of p must be tuned by "trial and error" to get the best trade-off between resolution and smoothness of the solution. The geometry of the tide gauge network, with no stations SW of the source area, decreases the resolution of the solution in that direction and increases the sensitivity towards changes in the smoothness parameter

p.

Figure 5 shows the solutions for the same set of parameters used in the "chessboard" test. In all cases the maxima of the initial displacement field correspond to segment BC (see figure 3), matching the 10-minute error contour of the BRT and the earthquake epicentre. The lateral extent of the source is roughly 160 km.

Figure 6 depicts the comparison between the forward computation using the initial displacement field

depicted in figure 5 for the preferred solution (p=0.3, r=8) and the observed data.

### 5. Discussion and Conclusions

The seismological data of the 25th November 1941 earthquake lead to an epicentre location at $-19.038^0$E, 37.4050N, and a moment magnitude of 8.3. The new epicentre lies west of the epicentre presented by

Guttenberg and Richter (1949). However, it is important to note that both determinations lie along segment BC (see figure 3) of the plate boundary whereas Udias et al. (1976) and Antunes (1944) locate the epicentre further north.

The analysis of the tsunami data shows a clear signal of small amplitude in most of the stations with an apparent change in the frequency content except for Leixões. The spectral analysis of the tsunami records

shows different frequency contents. These differences are mainly caused by differences in local morphological conditions and to the relative position of the station and the seismic source. Madeira station located approximately across strike shows the smallest dominant period of 6 minutes. Leixões, located along strike, shows the largest dominant period of 20 minutes. Ponta Delgada, Cascais and Lagos show dominant periods close to 11 minutes (see figures 1 and 2). These results are consistent with a tsunami

source located along the Nubia-Eurasia plate boundary.



The BRT simulation locate the minimum of the averaged travel time (7.2 minutes), at -19.000E, 37.350N quite close to the epicentre location. This result is consistent with the fact that the earthquake's co-seismic deformation induced the tsunami. It also shows the coherence of the tsunami dataset.

Because of the limitations inherent to the use of the linear shallow water approximation and the location of the tide gauges inside the harbours we opted to give different weight to the data points corresponding to up the first incoming wave. The set of solutions shown in figure 5, corresponding to different values of p (smoothness) and rmax (influence area), show a similar deformation pattern. All solutions show a smooth profile with a maximum upward displacement of +0.6 m compatible with the initial sea surface displacement computed with Okada (1985) formulae for the co-seismic deformation due to a fault in an 225 elastic half-space (see figure 5a). Moreover, all inverse-problem solutions favour the conclusion that the earthquake ruptured approximately 160 km of the plate boundary between $-20.249^0$E and $-18.630^0$E.

The results presented in figure 6 show that the inversion technique used here can fairly reproduce the first incoming wave for most of the observations presented in Table 1. In Cascais, the synthetic wave arrives eight minutes earlier than the observed wave. The location of the tide station inside the old harbour in a 230 very shallow area (less than 4 m) might be responsible for this discrepancy. Also, the smoothness coefficient used to compute the inverse solution may contribute to "spread" the lateral extent of the source thus producing earlier arrivals at some points on the coast. The observation in Essaouira is incompatible with all solutions presented in figure 5. The lack of the original tide records of Morocco and the poor quality of their reproduction in Debrach (1946) does not allow for further investigation. However, we can not exclude 235 the hypothesis of a local second tsunami source close to the coast of Morocco. Debrach (1946) and Rothé (1951) report damages to the submarine cables Brest–Casablanca and Brest–Dakar after the earthquake. This fact may suggest the occurrence of a submarine landslide close to the coast of Morocco.

In spite of the limitations inherent to the use of old instrumental records, some of them with a low-amplitude signal to noise ratio, the inversion of the tsunami waveforms provides an independent estimate of the 240 extension of the tsunami source and indirectly, the extent of the seismic source.

The seismic source solution (strike 76º, rake 161º) also favours the conclusion that the earthquake ruptured segment BC (see figure 3) of the plate boundary. This position coincides with a segment of the plate boundary that is marked by a change in strike when compared to the Gloria Fault between $-23.960^0$E and $-20.249^0$E (AB in figure 3), and to the eastern segment CD (see figure 3) between $-18.142^0$E and $-15.972^0$E. 245 While the relative motion between Eurasia and Nubia generates pure strike slip events along AB and CD, there is a significant thrust component associated with the seismic rupture within segment BC. This is consistent with both (seismic and tsunami) studies. Segments AB and CD follow the small circle of the Eurasia Nubia rotation pole while segment BC makes an angle of approximately $12^0$ with the small circle of Nuvel 1A.

250 Due to the small Eurasia-Nubia relative motion in this segment of the plate boundary (5 mm/yr), the recurrence of similar events is long. However, other parts of the same plate boundary can generate similar events, with noticeable effects in the North Atlantic coasts, and the tsunamigenic potential of the Gloria fault is relevant for the quantification of tsunami hazard in the North East Atlantic basin.



### 6. Acknowledgements

This work received funding from ASTARTE project Assessment Strategy and Risk Reduction for Tsunamis in Europe – Grant 603839 FP7. The authors wish to thank the fruitful discussions with Nuno Lourenço about the geology of the Gloria fault and the reviewers for their suggestions to improve this paper. The authors also thank Direção Geral do Território for making available data of Cascais and Lagos, Portugal. Copies of the paper tide records can be obtained from the authors.

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





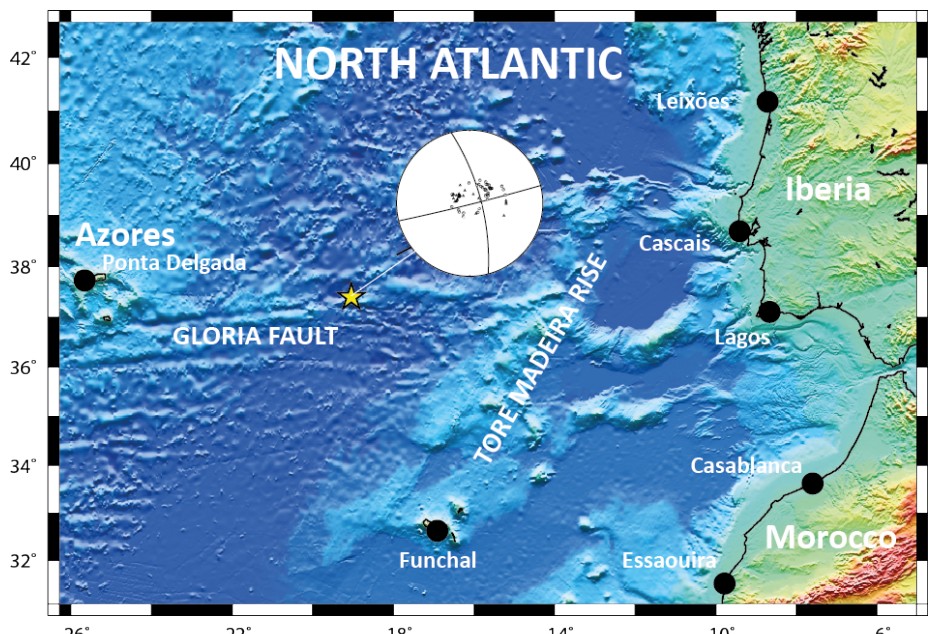

**Figure 1: General overview of the North East Atlantic at the latitude of Iberia and focal mechanism of the earthquake. The yellow star represents the epicenter of the earthquake. Black dots show the location of the tide stations.**





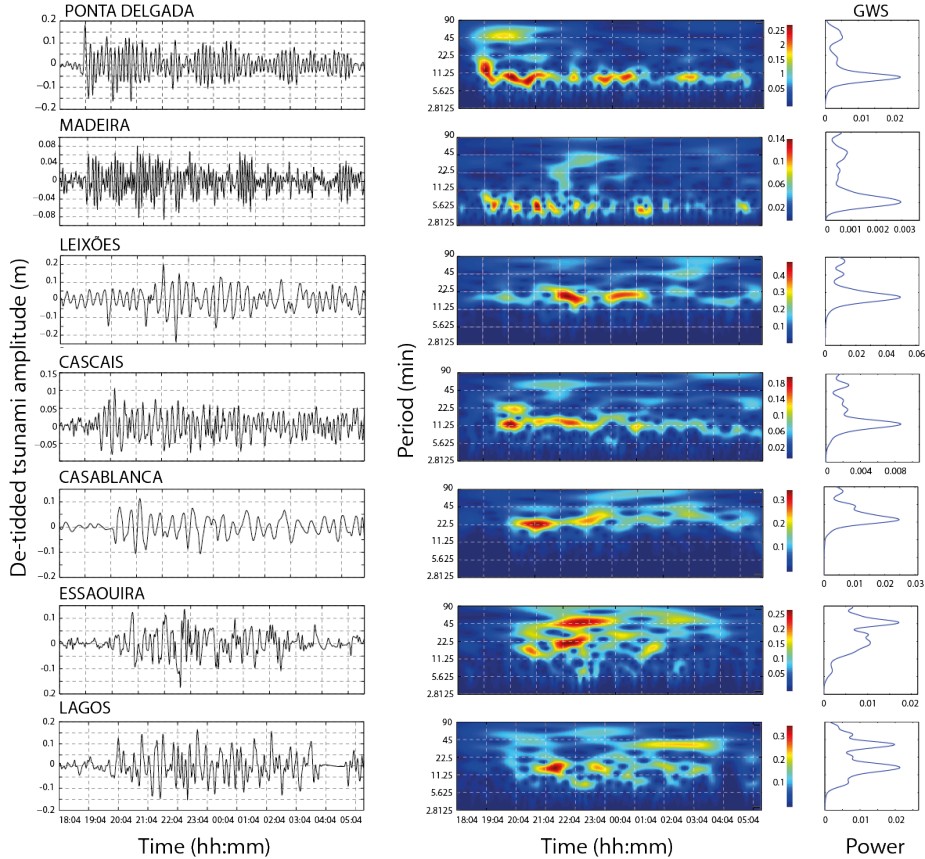

**Figure 2 For each station: Left: Tsunami signals (de-tided and filtered record), Middle: wavelet amplitude spectrum and Right: global wavelet spectrum (GWS) Initial time 18.04 hours is the time of the earthquake.**




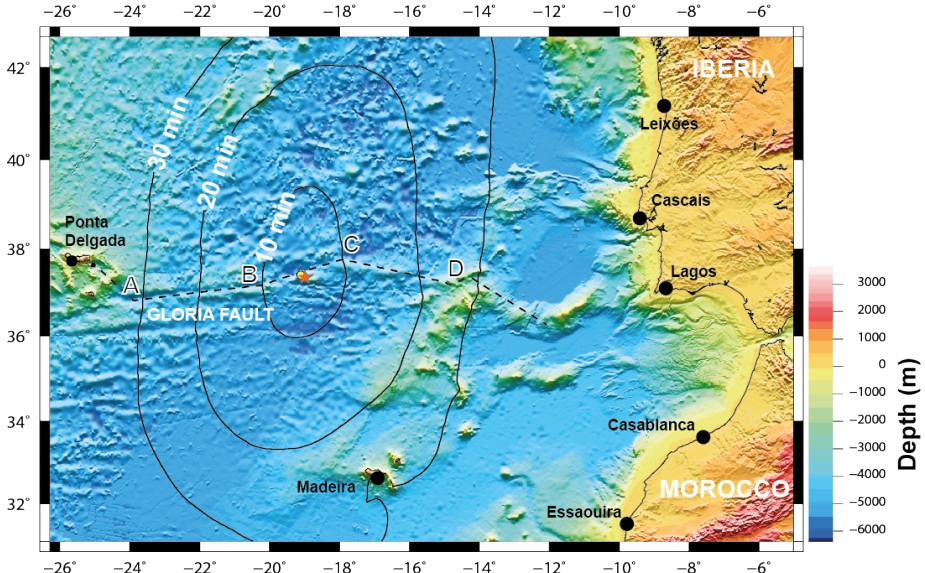

**Figure 3: Tsunami source location given by BRT. Red star represents the minimum of the backward ray tracing misfit (~7 min) at -19$^0$E, 37.35$^0$N less than 10 km away to the SE of the seismic epicentre -19.038$^0$E, 37.405$^0$N (yellow circle). Black dots represent the tide stations. Segments AB and CD follow the small circle of the Eurasia Nubia rotation pole; segment BC makes an angle of approximately 12$^0$ with the small circle of Nuvel 1A.**



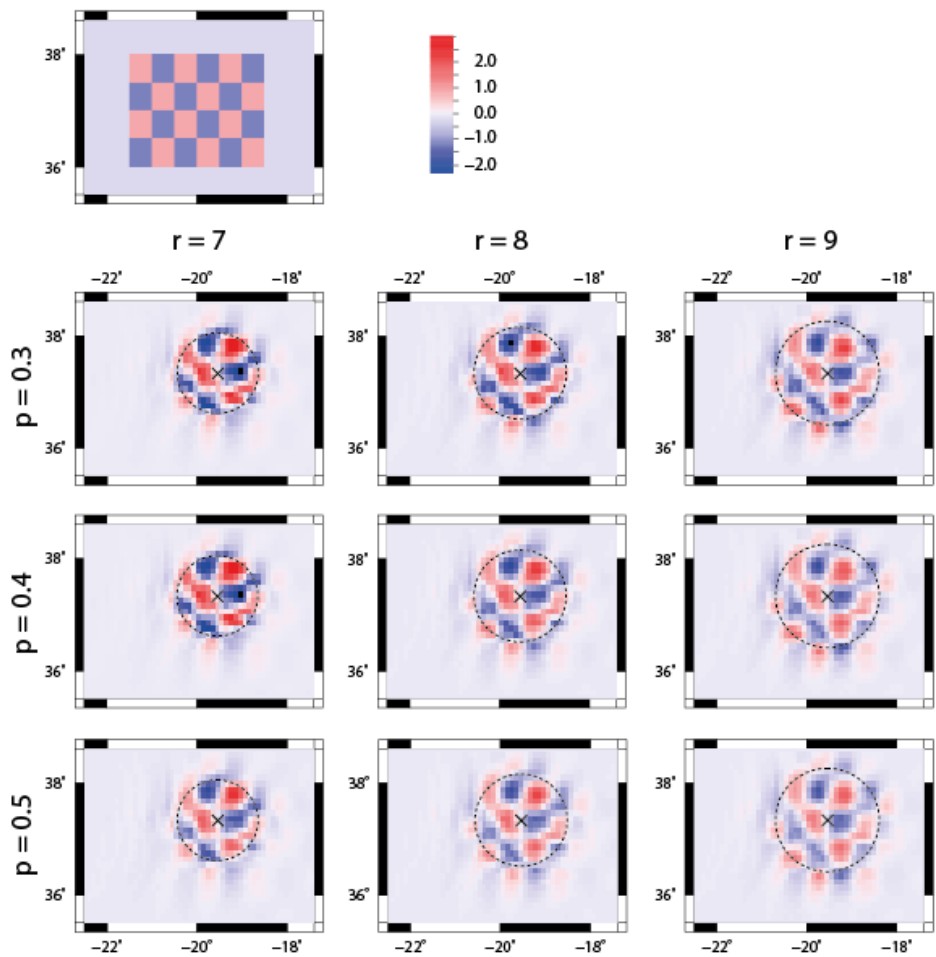

**Figure 4: Chessboard test of the inversion approach. The source has alternating +1m/-1m displacements and the synthetic waveforms computed with a shallow water code to the locations that correspond to our tide gauge network. Inversion results for different values of rmax (equation 4) and p (equation 5) are depicted.**




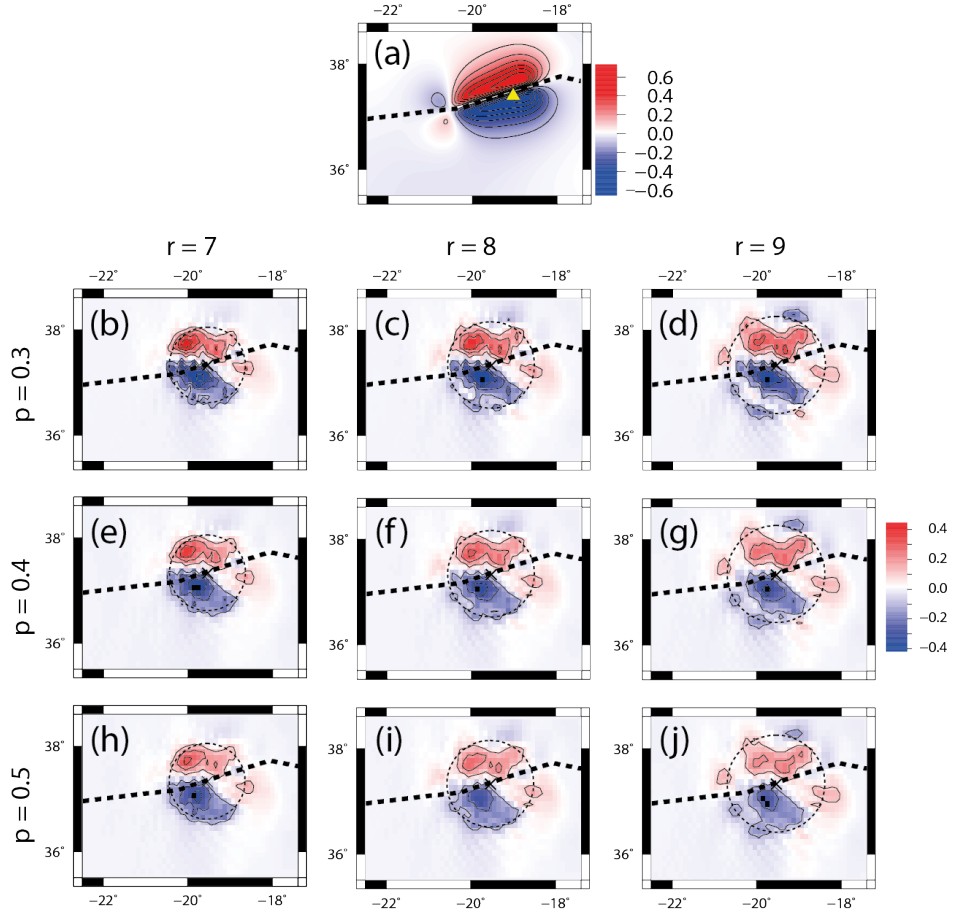

**Figure 5: (a) Co-seismic deformation using the focal mechanism parameters proposed in this study: L= 170 km, W=45 km; Strike=255$^0$, Rake=161$^0$, Dip=88$^0$, slip=8 m, yellow triangle depicts epicenter; (b)-(j) Initial displacement field of the 1941 Gloria Fault tsunami, obtained from waveform inversion for different values of r and p parameters, dashed line sketches the approximate location of the plate boundary.**





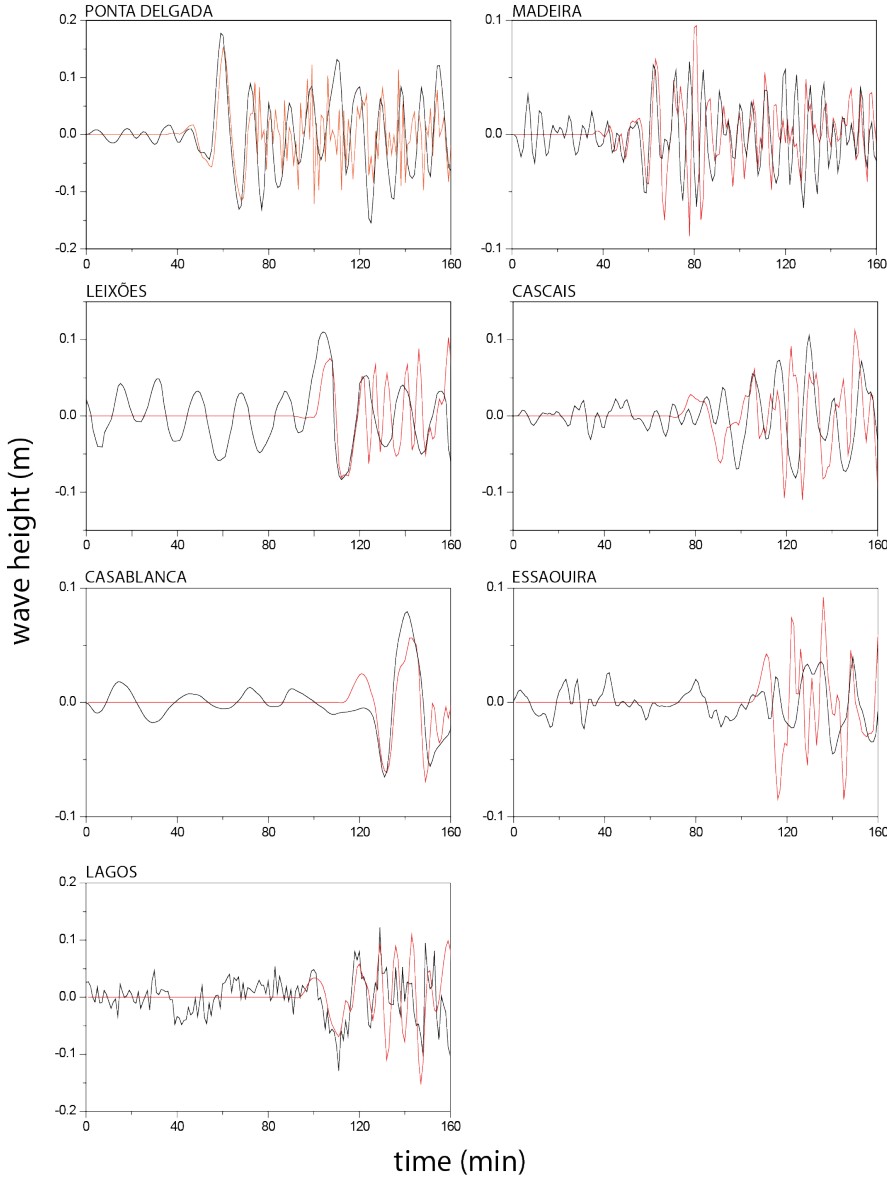

**Figure 6: Comparison between observed and synthetic waveforms. The solution shown corresponds to r=8 p=03. Red Line depicts the synthetic waveform. Black line depicts the observed waveform. Time zero is the time of the earthquake. Time 0 corresponds to the time of the earthquake, that is 18:03:54 of 25.11.1941.**

395





| Station | Longitude ($^0$E) | Latitude ($^0$N) | Depth (m) | Max. Amplitude (m) | Tsunami Travel Time (minutes) |
|---|---|---|---|---|---|
| Ponta Delgada | -25.65 | 37.73 | 49.4 | 0.20 | 53.0 |
| Madeira | -16.91 | 32.63 | 246.9 | 0. 08 | 55.0 |
| Leixões | -8.71 | 41.17 | 10.1 | 0.20 | 94.0 |
| Cascais | -9.41 | 38.69 | 11.5 | 0.11 | 92.0 |
| Casablanca | -7.59 | 33.62 | 13.0 | 0.09 | 120.0 |
| Essaouira | -9.78 | 31.51 | 9.2 | 0.12 | 110.0 |
| Lagos | -8.66 | 37.10 | 7.8 | 0.12 | 107.0 |

400 **Table 1. POIs of the 1941 tsunami used in this study. Longitude, Latitude and Depth refer to the Point of Interest. POIs are named after the closest tide station.**