# Peer review of "New Study on the 1941 Gloria Fault Earthquake and Tsunami"

_Natural Hazards and Earth System Sciences, 2016_

## Referee Comment (RC1) · Anonymous Referee #1 · 5 Jun 2016

The NHESS paper "New Study on the 1941 Gloria Fault Earthquake and Tsunami" by Baptista et al. provides a new interpretation of this important historical earthquake by analyzing seismograms and tide gauge records. The tide gauge records, in particular, are inverted in terms of both travel-times and waveforms, using state-of-the art techniques. This study places the earthquake in a new location compared to several other older analyses, although on the same section of the Gloria fault as the analysis of Gutenberg and Richter (1949).

This is a straightforward and clearly presented study. I only have three general, substantive comments indicated below:

- The technique to locate the epicenter from the seismograms needs more description. In particular, what is the method behind ""Hypocenter" code running under SEISAN

environment" (P60)? In addition, it would be very useful to the reader if a waveform comparison of the seismograms is presented in a figure. Displaying the newly-acquired historic seismograms would also be very interesting.

- Could the significant topography surrounding the Gloria fault also play a role in tsunami generation, in terms of horizontal movement? This was discussed by Ishii et al. [2013] in an analogous study of the 2012 Mw 8.6 Indian Ocean event, with regard to displacement of the Ninety-East Ridge.

- Is the apparent rotation of the inverted displacement field shown in Fig. 5 caused by the azimuthal distribution of tide gauge stations or some other artifact of the inversion?

Specific technical comments are indicated below:

- It would be helpful to show the fault trace for the Gloria fault on Fig. 1, as well as how it is segmented. Also, I cannot discern the 1st motion polarities on the focal mechanism in Fig. 1 (e.g., the "BC segment").

- P45: "...new set of old seismograms" better phrased as "...newly acquired set of historic seismograms".

- P65: What is the Dineva et al. (2002) approach? Please describe in the manuscript.

Reference

Ishii, M., E. Kiser, and E. L. Geist (2013), Mw 8.6 Sumatran earthquake of 11 April 2012: Rare seaward expression of oblique subduction, Geology, 41, 319-322.
* * *

---

## Referee Comment (RC2) · Anonymous Referee #2 · 4 Jul 2016

This manuscripts is on the a re-evaluation of the epicenter location of the M∼8.3-8.4 25th November 1941 in the North East (NE) Atlantic basin, occurred along the Eurasia-Nubia plate boundary between the Azores and the Strait of Gibraltar as one of the largest submarine strike-slip earthquakes ever recorded in the region, using seismological data not included in previous studies. Furthermore, the authors inverted recorded tsunami waveforms to infer the initial sea surface displacement using Empirical Green Functions without prior assumptions on the geometry of the source to verify the re-location. The study attempts to show that the tsunami was generated due to earthquake's co-seismic deformation but the authors cannot exclude the hypothesis of a local second tsunami source close to the coast of Morocco. The manuscript is clearly written and concise. The study is limited by the use of old instrumental records, where some of them with a low-amplitude, which is a common limitation in the analysis of

historical tsunamis. The manuscript inarguably addresses relevant scientific questions within the scope of NHESS. It presents both new data and makes effective use of combining several earlier proposed methods. They are up to international standards and both the assumptions and limitations of the used methodologies are clearly written. Since this is the first time that the associated tsunami has been analysed comprehensively, the study should be considered a contribution to the evaluation of tsunami hazard in the North East Atlantic basin. Yet, the question remains: was the tsunami due to the earthquake's co-seismic deformation or was there a submarine landslide close to the coast of Morocco?

---

## Author Comment (AC1) · 7 Jul 2016

Anonymous Referee #1, comment 1 (0) page 2, line 60 (1) comments from referees/public: The technique to locate the epicenter from the seismograms needs more description. In particular, what is the method behind ""Hypocenter" code running under SEISAN environment" (P60)? (2) author's response: We accept the comment. (3) author's changes in manuscript: the paragraph reads now: "To compute the hypocenter, we used "Hypocenter", a damped least square algorithm for earthquake location (Lienert et al., 1986), running under SEISAN environment, a seismic analysis package containing a complete set of programs and a simple database for analyzing earthquake data (Ottemöller et al., 2011). The reference Lienert et al., 1986 was also added to the reference list.

[Figure]

Anonymous Referee #1, comment 2 (0) page 2, line 61 (1) comments from referees/public: In addition, it would be very useful to the reader if a waveform comparison of the seismograms is presented in a figure. Displaying the newly-acquired historic seismograms would also be very interesting. (2) author's response: We accept the comment. (3) author's changes in manuscript: A new figure 2 was added to the manuscript. All other figures were renumbered accordingly.

Anonymous Referee #1, comment 3 (0) page 7, line 224-225 (1) comments from referees/public: Could the significant topography surrounding the Gloria fault also play a role in tsunami generation, in terms of horizontal movement? This was discussed by Ishii et al. [2013] in an analogous study of the 2012 Mw 8.6 Indian Ocean event, with regard to displacement of the Ninety-East Ridge. (2) author's response: The comparison shown in figure 6 (now figure 7) was made considering only the vertical component of the co-seismic displacement constrained by the horizontal resolution of the Green Function Database. The role of the horizontal motion of the seafloor close to the source can be a good guess for the interpretation of the solution, due to the sharp vertical offset associated with the Gloria Fault, but is outside the scope of the paper. (3) author's changes in manuscript: no change.

Anonymous Referee #1, comment 4 (0) page 7, line 230-231 (1) comments from referees/public: is the apparent rotation of the inverted displacement field shown in Fig. 5 caused by the azimuthal distribution of tide gauge stations or some other artifact of the inversion? (2) author's response: We cannot attribute the apparent rotation of the solutions displayed in figure 5 to the azimuthal distribution of the tide gauge stations as the "chessboard test" shows (figure 4). This rotation may result from the the apparent "contradiction" between the records from Cascais (see lines 230-231) and Lagos tide stations, but we have no independent assessment of this problem to weight differently the station data. Another possible explanation is the effect of a secondary tsunami source – a possible submarine landslide (see answer to referee 2). (3) author's changes in manuscript: no change.

Anonymous Referee #1, comment 5 (0) page 11, line 369 (1) comments from referees/public: It would be helpful to show the fault trace for the Gloria fault on Fig. 1, as well as how it is segmented. (2) author's response: The segmentation of Gloria fault is shown in figure 4. In figure 1 it would jeopardize the direct analysis of bathymetric data by the readers. (3) author's changes in manuscript: No action.

Anonymous Referee #1, comment 6 (0) figure 1 (1) comments from referees/public: I cannot discern the 1st motion polarities on the focal mechanism in Fig. 1 (e.g., the "BC segment"). (2) author's response: We accept the comment. (3) author's changes in manuscript: Figure 1 was edited accordingly.

Anonymous Referee #1, comment 7 (0) page 2, line: 45 (1) comments from referees/public: ". . .new set of old seismograms" better phrased as ". . .newly acquired set of historic seismograms" (2) author's response: We accept the comment. (3) author's changes in manuscript: The manuscript was changed accordingly.

Anonymous Referee #1, comment 8 (0) page 2, line: 69-70 (1) comments from referees/public: P65: What is the Dineva et al. (2002) approach? please describe in the manuscript. (2) author's response: We accept the comment and the sentence was rephrased and completed. (3) author's changes in manuscript: The paragraph was changed to "We computed the scalar seismic moment using the (Dineva et al., 2002) approach. Original analogue seismograms were digitized and the seismic moment is computed from the spectra of body waves ground motion independently for each component. Twenty-six seismograms from fourteen seismic stations were digitized."

The revised version of the manuscript is uploaded named Revised_discussion_paper_new_study_1941_Atlantic_tsunami with changes highlighted in yellow

Please also note the supplement to this comment:
http://www.nat-hazards-earth-syst-sci-discuss.net/nhess-2016-130/nhess-2016-130-

AC1-supplement.pdf

[Figure]

**Supplement:**

**New Study on the 1941 Gloria Fault Earthquake and Tsunami**

Maria Ana Baptista1,2, Jorge Miguel Miranda2,3, Josep Batlló4, Filipe Lisboa3, Joaquim Luis2,5, Ramon Maciá6

[revised manuscript text omitted]

- In this study, we present a relocation of the earthquake source using the phases published by the ISC bulletin complemented with readings from nearby stations AVE in Morocco (-7.41330E, 33.29610N), FBR in Spain (2.12390E, 41.41840N) and MAL in Spain (-4.42920E, 36.7620N). To compute the hypocenter, we used "Hypocenter", a damped least square algorithm for earthquake location (Lienert et al., 1986), running under SEISAN environment, a seismic analysis package containing a complete set of programs and a simple database for analyzing earthquake data (Ottemöller et al., 2011).
- Additionally, we tested two velocity models to compute the epicenter. The IASPEI91 velocity model (Kennett et al., 1991) locates the epicenter at -18.965oE, 37.344oN. The regional velocity model by Carrilho et al. (2004) for the Portuguese area locates the epicenter at -19.038oE, 37.405oN, which is our preferred solution. The focal depths given by these models are 12km (rms=2.5s) and 5.3km (rms=2.2s) respectively.

We computed the scalar seismic moment using the (Dineva et al., 2002) approach. Original analogue
seismograms were digitized and the seismic moment is computed from the spectra of body waves ground motion independently for each component. Twenty-six seismograms from fourteen seismic stations were digitized. Using P waves from thirteen stations we obtain a seismic moment of 3.96 1021 Nm, while using S waves from seven stations the seismic moment gives 2.96 x1021 Nm. These seismic moment values correspond to a moment magnitude of 8.3±0.2 and 8.2±0.3 respectively. To re-evaluate the focal

75 mechanism, we used the polarities from seventy-two seismograms. The parameters of the focal mechanism

are strike  $76^{\circ}$ , dip  $88^{\circ}$  and rake angle  $-161^{\circ}$ . The focal mechanism is depicted in figure 1 and an example of a newly digitized seismogram is shown in figure 2.

**3. Tsunami data analysis**

[revised manuscript text omitted]

$$\eta_{\rm m}(t_{\rm t}) = \sum_{\rm i=1}^{\rm nl} \sum_{\rm j=1}^{\rm nc} G_{\rm ij}^{\rm m}(t_{\rm t}) h_{\rm ij} \tag{4}$$

where (nl x nc) are the set of unit sources and  $h_{ij}$  is the amplitude of the initial water displacement attributed to the ij unit source.  $G_{ij}^{m}(t_t)$  is the response to the unitary source of the ith element of the sea surface at the jth location and m is the total number of unitary sources. The waveform inversion is simply obtained by solving:

$$\begin{bmatrix} G_{11}^{1}(t_{1}) & G_{12}^{1}(t_{1}) & G_{13}^{1}(t_{1}) & \dots & G_{nl nc}^{1}(t_{1}) \\ & \dots & & \\ G_{11}^{1}(t_{t}) & G_{12}^{1}(t_{t}) & G_{13}^{1}(t_{t}) & \dots & G_{nl nc}^{1}(t_{t}) \\ & \dots & & \\ G_{nl nc}^{m}(t_{1}) & G_{12}^{m}(t_{1}) & G_{13}^{m}(t_{1}) & \dots & G_{nl nc}^{m}(t_{1}) \\ & \dots & & \\ G_{nl nc}^{m}(t_{t}) & G_{12}^{m}(t_{t}) & G_{13}^{m}(t_{t}) & \dots & G_{nl nc}^{m}(t_{t}) \end{bmatrix} \begin{bmatrix} h_{11} \\ h_{12} \\ h_{13} \\ \dots \\ h_{nl nc} \end{bmatrix} = \begin{bmatrix} \eta_{1}(t_{1}) \\ \dots \\ \eta_{1}(t_{t}) \\ \dots \\ \eta_{m}(t_{1}) \\ \dots \\ \eta_{m}(t_{t}) \end{bmatrix}$$
(5)

160

155

We incorporate the suggestion by Tsushima et al. (2009) defining an influence area around the minimum of the BRT. The initial water displacement is forced to become null as the distance to the minimum increases, therefore allowing the computation of a spatially compact solution. To do this, we added to (5) additional equations of the form:

$$\begin{bmatrix} 0 & 0 & 0 & \dots & q & \dots & 0 \end{bmatrix} \begin{bmatrix} h_{11} \\ h_{12} \\ h_{13} \\ \dots \\ h_{nl nc} \end{bmatrix} = \begin{bmatrix} 0 \end{bmatrix}$$
(6)

where q is a positive number given by:

165
$$q(r) = 10 * (\frac{r}{r_{max}} - 1)$$
 (7)

and  $r > r_{max}$  is the distance between each parameter (unit source) and the minimum of BRT.  $r_{max}$  is a cut-off distance after which the initial water displacement converges to zero. Equations (5) and (6) form a mixed-determined linear inversion problem (Menke, 2012) which can be solved in the sense of the least squares weighted minimum norm, by;

170
$$h^{\text{est}} = [G^{\text{T}}W_{\text{e}}G + p^{2}D^{\text{T}}D]^{-1}G^{\text{T}}W_{\text{
[revised manuscript text omitted]

325 10.1038/237217a0, 1972.

305

315

Lienert, B. R., Berg, E. and Frazer, L. N.: Hypocenter: an Earthquake Location Method using Centered, Scaled and Adaptively Damped Least Squares, Bull. Seis. Soc. Am., 73(3), 771-783, doi: , 1986) Luis, J. F.: Mirone: A multi-purpose tool for exploring grid data. Computers & Geosciences, 33(1), 31-41, doi: 10.1016/j.cageo.2006.05.005, 2007.

Luis, J. F., and Miranda, J. M.: Reevaluation of magnetic chrons in the North Atlantic between 35 N and
 47 N: Implications for the formation of the Azores Triple Junction and associated plateau. J. of Geophys.
 Res., Solid Earth, 113(B10), doi :10.1029/2007JB005573, 2008.

Lynnes, C.S., and Ruff, L.J.: Source process and tectonic implications of the great 1975 North Atlantic earthquake. Geophys. J. Int., 82, 3, 497-510, doi: 10.1111/j.1365-246X.1985.tb05148.x, 1985.

335 Menke, W.: Geophysical data analysis: Discrete inverse theory. Third Edition, pp 293, ISBN 978-0-12-397160-9. Academic Press, 2012.

Miranda, J. M., Baptista, M. A., and Omira, R.: On the use of Green's summation for tsunami waveform estimation: a case study. Geophys. J. Int., 199(1), 459-464, doi: 10.1093/gji/ggu266, 2014.

Moreira, V.S.: Tsunamis observados em Portugal. Publicacao GEO,134 (in Portuguese), 1968.

340 Ottemöller, L., Voss, P., and Havskov, J.: Seisan earthquake analysis software for Windows, Solaris, Linux and MacOsX. Dept. Earth Sci., Univ. Bergen, Bergen, Norway, 2011.

Rothé, J. P.: The structure of the bed of the Atlantic Ocean. Eos, Transactions American Geophysical Union, 32(3), 457-461, doi: 10.1029/TR032i003p00457, 1951.

Satake, K.: Inversion of tsunami waveforms for the estimation of a fault heterogeneity: Method and numerical experiments. J. of Physics of the Earth, 35(3), 241-254, doi:org/10.4294/jpe1952.35.24, 1987.

345

Satake, K.: Depth distribution of co-seismic slip along the Nankai Trough, Japan, from joint inversion of geodetic and tsunami data. J. of Geophys. Res.: Solid Earth (1978–2012), 98(B3), 4553-4565, doi: 10.1029/92JB01553, 1993.

Satake, K., Baba, T., Hirata, K., Iwasaki, S., Kato, T., Koshimura, S., Takenaka, J. and Terada, Y.: Tsunami
source of the 2004 off the Kii Peninsula earthquakes inferred from offshore tsunami and coastal tide gauges.
Earth, Planets, and Space, 57(3), 173-178, doi: 10.1186/BF0335181, 2005.

Titov, V. V., Gonzalez, F. I., Bernard, E. N., Eble, M. C., Mofjeld, H. O., Newman, J. C., and Venturato, A. J.: Real-time tsunami forecasting: Challenges and solutions. In Developing Tsunami-Resilient Communities, 41-58, Springer Netherlands, doi: 10.1007/1-4020-3607-8\_3, 2005.

355 Tsushima, H., Hino, R., Fujimoto, H., Tanioka, Y. and Imamura, F.: Near- field tsunami forecasting from cabled ocean bottom pressure data. J. of Geophys. Res.: Solid Earth (1978–2012), 114(B6), doi:10.1029/2008JB005988, 2009.

Udias, A., Arroyo, A. L., and Mezcua, J.: Seismotectonic of the Azores-Alboran region. Tectonophysics, 31(3), 259-289, doi:10.1016/0040-1951(76)90121-9, 1986.

Wu, T. R., and Ho, T. C.: High resolution tsunami inversion for 2010 Chile earthquake. Nat. Hazards and Earth Syst. Sci., 11(12), 3251-3261, doi:10.5194/nhess-11-3251-2011, 2011.

Yasuda, T., and Mase, H.: Real-time tsunami prediction by inversion method using offshore observed GPS buoy data: nankaido. J. of Waterway, Port, Coastal, and Ocean Eng., 139(3), 221-231, doi:10.1061/(ASCE)WW.1943-5460.0000159, 2012.

Figure 1: General overview of the North East Atlantic at the latitude of Iberia and focal mechanism of the earthquake. The yellow star represents the epicenter of the earthquake. Black dots show the location of the tide stations.

---

## Author Comment (AC2) · 7 Jul 2016

(1) comments from referees/public: This manuscript is on the re-evaluation of the epicenter location of the M$\hat{a}\acute{L}$ij8.3- 8.4 25th November 1941 in the North East (NE) Atlantic basin, occurred along the Eurasia-Nubia plate boundary between the Azores and the Strait of Gibraltar as one of the largest submarine strike-slip earthquakes ever recorded in the region, using seismological data not included in previous studies. Furthermore, the authors inverted recorded tsunami waveforms to infer the initial sea surface displacement using Empirical Green Functions without prior assumptions on the geometry of the source to verify the re-location. The study attempts to show that the tsunami was generated due to earthquake's co-seismic deformation but the authors cannot exclude the hypothesis of a local second tsunami source close to the coast of Morocco. The manuscript is clearly written and concise. The study is limited by the use of old

instrumental records, where some of them with a low-amplitude, which is a common limitation in the analysis of C1 NHESSD Interactive comment Printer-friendly version Discussion paper historical tsunamis. The manuscript inarguably addresses relevant scientific questions within the scope of NHESS. It presents both new data and makes effective use of combining several earlier proposed methods. They are up to international standards and both the assumptions and limitations of the used methodologies are clearly written. Since this is the first time that the associated tsunami has been analysed comprehensively, the study should be considered a contribution to the evaluation of tsunami hazard in the North East Atlantic basin. Yet, the question remains: was the tsunami due to the earthquake's co-seismic deformation or was there a submarine landslide close to the coast of Morocco?

(2) author's response: We can not discard the fact that a local landslide occurred because the break of the submarine cables is documented. We have no data about the exact location of the submarine cable neither where it was broken. This lack of information makes the modelling of the landslide very difficult. We think that the occurrence of a landslide close to the coast of Africa (Morocco and Senegal) might influence the signal of the Morocco tide records. However, our strongest tsunami observation is in the Azores (Ponta Delgada); this station is too far away from the submarine cable (and to the possible landslide location) therefore we conclude that the tsunami observed there is due to the earthquake's co-seismic deformation. (3) author's changes in manuscript: no change.

The revised manuscript according to the requirements of referee 1 can be found in the supplement

Please also note the supplement to this comment:
http://www.nat-hazards-earth-syst-sci-discuss.net/nhess-2016-130/nhess-2016-130-AC2-supplement.pdf